# Behavioural factors associated with fear of litigation as a driver for the increased use of caesarean sections: a scoping review

Sarah Elaraby,[1,2] Elena Altieri [ID],[2] Soo Downe,[3] Joanna Erdman,[4] Sunny Mannava,[5] Gill Moncrieff,[3] B R Shamanna,[5] Maria Regina Torloni [ID],[6] Ana Pilar Betran [ID][7]

For numbered affiliations see end of article.

**Correspondence to**
Dr Ana Pilar Betran;
betrana@who.int

## ABSTRACT

**Objective** To explore the behavioural drivers of fear of litigation among healthcare providers influencing caesarean section (CS) rates.

**Design** Scoping review.

**Data sources** We searched MEDLINE, Scopus and WHO Global Index (1 January 2001 to 9 March 2022).

**Data extraction and synthesis** Data were extracted using a form specifically designed for this review and we conducted content analysis using textual coding for relevant themes. We used the WHO principles for the adoption of a behavioural science perspective in public health developed by the WHO Technical Advisory Group for Behavioural Sciences and Insights to organise and analyse the findings. We used a narrative approach to summarise the findings.

**Results** We screened 2968 citations and 56 were included. Reviewed articles did not use a standard measure of influence of fear of litigation on provider's behaviour. None of the studies used a clear theoretical framework to discuss the behavioural drivers of fear of litigation. We identified 12 drivers under the three domains of the WHO principles: (1) cognitive drivers: availability bias, ambiguity aversion, relative risk bias, commission bias and loss aversion bias; (2) social and cultural drivers: patient pressure, social norms and blame culture and (3) environmental drivers: legal, insurance, medical and professional, and media. Cognitive biases were the most discussed drivers of fear of litigation, followed by legal environment and patient pressure.

**Conclusions** Despite the lack of consensus on a definition or measurement, we found that fear of litigation as a driver for rising CS rates results from a complex interaction between cognitive, social and environmental drivers. Many of our findings were transferable across geographical and practice settings. Behavioural interventions that consider these drivers are crucial to address the fear of litigation as part of strategies to reduce CS.

## INTRODUCTION

Over the past decades, rates of caesarean section (CS) have been rising steadily worldwide, in many cases without medical indication.[1] The latest available estimates show that worldwide the average CS rate increased from 6.7% in 1990 to 21% in 2018.[2] The medically unnecessary use of CS impacts the quality of care and is a burden on often stretched healthcare systems,[3] in addition to exposing women and offspring to the short-term and long-term risks associated with this surgery.[4 5]

Medical litigation can be defined as a legal dispute that involved carrying out a lawsuit or civil action against healthcare providers or a medical entity.[6] We use litigation here as an umbrella term to cover a variety of lawsuits including medical malpractice, negligence and the associated legal liability, risk and settlements.

Fear of litigation (FoL) has been commonly recognised as an influential factor in medical decision making.[7] Obstetrics is one of the leading medical specialities in terms of litigation risk and cost.[8–11] FoL and defensive

## STRENGTHS AND LIMITATIONS OF THIS STUDY

⇒ We systematically explore global accounts of fear of litigation (FoL) and associated behavioural factors as drivers for the increased use of caesarean section (CS) providing a broad overview of the evidence and identifying research gaps.

⇒ We employed a behavioural approach and the WHO behavioural principles which allowed us to classify drivers within specific domains (cognitive, social and environmental) offering clear pathways for the design of interventions to address these fears.

⇒ We developed a comprehensive search strategy and searched multiple databases but we did not search grey literature and other informal publications.

⇒ While our interest was understanding the drivers of medically unnecessary CS, the studies did not differentiate between necessary and unnecessary CS; hence, we reported FoL in relation to 'increasing use of CS' rather than 'increasing use of unnecessary CS'.

medicine (practice wherein a healthcare professional makes decisions out of FoL and not for the benefit of the patients) are commonly cited reasons for performing CS,[12–14] sometimes without medical indication.[12 15 16] Beyond mode of birth (MoB) decision making, FoL has profound impact on maternity care, with studies across the world documenting high rates of defensive medicine,[17–22] a decline in the desire to practice obstetrics or early retirement of practicing obstetricians due to litigation concerns.[15 23]

Despite the frequent emphasis on FoL as a driver of increased CS, there is no agreed definition or a standardised tool to measure it, which reduces the ability to assess its role as a driver of CS rates. For the purpose of this review, we rely on explicit mentions of FoL and its consequences in relation to MoB decision making.

### Behavioural drivers of fear of litigation

Evidence from behavioural science indicates that our decisions and behaviours are often not deliberate and rational, but rather automatic and profoundly influenced by our environment.[12 24] Understanding human behaviour is essential to improve the design and implementation of interventions aiming to influence decision making of providers. To improve understanding of FoL in relation to MoB, and map existing evidence that could inform interventions to optimise the use of CS, we conducted a scoping review asking the following questions: (a) how is FoL defined by providers in relation to MoB? and (b) what are the various behavioural drivers influencing FoL and affecting decision making for MoB?

We used the principles for the adoption of a behavioural science perspective in public health developed by WHO as a guideline to organise and analyse the findings of the review.[25] Based on these behavioural principles, we mapped the different types of influences on behaviour, namely cognitive and psychological factors, social and cultural factors, and environmental factors driving FoL among providers. The findings of this review further informed this initial mapping.

## METHODS

The scoping review followed the standard methodology recommended by the Joanna Briggs Institute and is reported in adherence to Preferred Reporting Items for Systematic Reviews and Meta-Analyses for Scoping reviews.[26 27]

### Data sources and searches

We searched three electronic databases (MEDLINE via PubMed, Scopus and WHO Global Index) for studies published from 1 January 2001 to 9 March 2022. The search strategy used two main concepts (and their synonyms): 'caesarean section' and 'litigation' (detailed search strategy in online supplemental file 1). We also screened the reference lists of reviews and included

articles for potentially relevant studies not captured by the electronic search.

### Selection criteria

The review included primary research assessing drivers of FoL experienced by providers in relation to MoB or decision making for CS. We included studies published after 2000 with qualitative, quantitative or mixed-method designs. Studies from any country or region were eligible for inclusion. Studies were included if they were published in English, Spanish, Portuguese or French. We excluded case reports or case series (n<6), reports not describing original research (eg, commentaries, communications, news articles, reviews and so on), or if they only covered pure legal analysis (ie, not reporting original data regarding decision making). Reviews were excluded but their reference list screened for eligible studies. We included articles covering maternity care by providers in any cadre and involved in any health setting or sector.

### Study identification and data extraction

Citations retrieved from electronic databases were uploaded into Covidence (Veritas Health Innovation, Melbourne, Australia) and duplicates were excluded. Titles and abstracts were screened in duplicate, and full texts of potentially relevant studies were obtained and assessed by two reviewers independently (SE and APB). Discrepancies were discussed until a consensus was reached.

A data-extraction form was specifically designed for this review to capture key aspects of the studies (setting, design, participants, behavioural drivers and factors affecting FoL; detailed variables extracted in the online

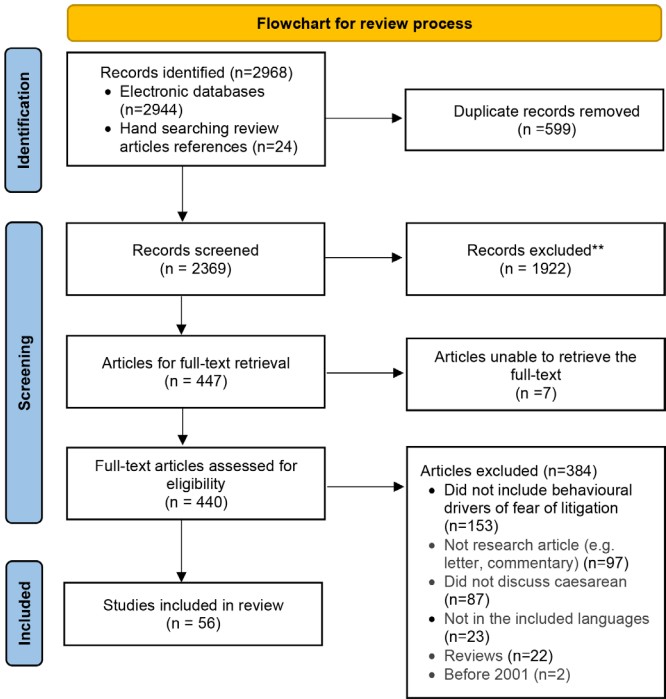

**Figure 1** Flowchart of process of study identification and inclusion.

**Table 1** Main characteristics of 56 studies included in the scoping review

| Characteristic | No. of studies (n=56) | References |
|---|---|---|
| Economic category (World Bank classification) | | |
| Low-income, lower-income or middle-income countries | 22 | 13 14 17 32 33 35 36 41 47 49–55 60 61 64 66–68 |
| High-income | 34 | 16 18 19 21 22 34 37–39 42–46 48 56–59 62 63 69–73 75–80 88 89 |
| Study design* | | |
| Quantitative | 33 | 13 14 16–19 21 22 33–36 38 39 42–46 54 55 61 62 70–73 77–80 88 89 |
| Qualitative | 17 | 32 37 47 48 50–53 56–59 64 68 69 75 76 |
| Mixed methods | 6 | 49 60 63 66 67 |
| Year of publication | | |
| 2001–2010 | 17 | 16 19 42–44 51 55–58 63 66 70 71 73 88 89 |
| 2011–2022 | 39 | 13 14 17 18 21 22 32–39 45–50 52–54 59 60 62 64 67–69 72 75–80 |
| Number of countries | | |
| 1 | 53 | 13 14 16–19 22 32–39 42–60 62–64 66–68 70 72 73 76–80 88 89 |
| >1 | 3 | 69 71 75 |
| Perspectives included* | | |
| Doctors | 47 | 13 14 16–19 21 32 34–38 41 47–64 66–73 75–80 88 89 |
| Midwives, nurses and doulas | 12 | 34 37 41 48 52 57 59 61 66 72 76 80 |
| Service users (pregnant women and their families) | 5 | 48 52 59 66 72 |
| Administrators and health service decision makers | 2 | 32 48 |
| Lawyers | 1 | 72 |
| Other (including records) | 9 | 22 33 39 42–46 48 67 |

*Articles can include multiple perspectives.

supplemental file 1). We used content analysis to identify emerging themes in the articles, assisted by Atlas.ti 9, for further analysis of FoL, its drivers and how they influence decision making regarding MoB.[28 29] We developed a qualitative codebook informed by previous relevant reviews,[12 24] and the behavioural principles described before.[30 31] One reviewer (SE) supervised by two reviewers (EA and APB) conducted the data extraction and developed the initial codebook. The codebook was iteratively modified as themes emerged inductively during the process of extraction of the included articles in this scoping review, and through continuous team discussions and reflection. We finally cut out redundant themes or those that were either outside the study scope or were included in only one study.

### Data synthesis
Using the qualitative coding process, along with the extraction sheet, it was possible to identify and compare different behavioural drivers of FoL within and across studies.[28 29] This started with applying the codebook and extraction sheet to a few studies, and then going through an iterative process, with continuous revision of codes and overarching themes as more studies were analysed. This process was conducted by one reviewer (SE) and discussions among co-authors informed the classification

of drivers according to the WHO principles and the definition of how they interact.

In this manuscript, 'litigation' and associated terms are used in a non-technical way reflecting their use in the studies reviewed which is often from the perspective of non-legally trained persons, namely providers. It also serves as a construct that covers the risk of the legal process, and the experience of litigation itself, rather than specific litigation outcomes.

### Patient and public involvement
Patients and/or the public were not involved in the design, or conduct, or reporting, or dissemination plans of this scoping review.

### RESULTS
A total of 2968 citations were identified (2944 from electronic databases and 24 from other sources). We selected 440 studies for full-text evaluation and included 56 in this review (figure 1).

### Main characteristics of included studies
Table 1 shows the main characteristics of the 56 included studies. Thirty-four (60.7%) were from high-income countries (HICs) while 22 (39.3%) were from low/

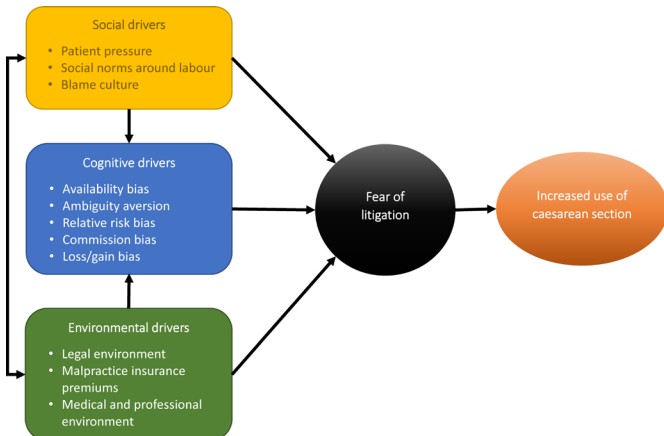

**Figure 2** Conceptual framework for the classification of drivers of fear of litigation under the three domains defined in the WHO principles.

middle-income countries (LMICs). Most (33, 58.9%) had quantitative designs and were published after 2010. All but three included data from a single country; 47 (83.9%) included the perspective of doctors, 12 (21.4%) of midwives, nurses or doulas, 5 (8.9%) of service users and 1 (1.8%) of lawyers. Study details including behavioural drivers reported for each of the 56 studies are presented in online supplemental file 2.

### Defining and measuring fear of litigation
There was marked heterogeneity in defining or describing FoL in the context of MoB decision making. Other terms used to imply FoL included liability concerns,[22 32–34] medico-legal fears[13 16 35 36] and defensive medicine.[14 17 18 21 33 37–40] The only tool we found was a risk attitude and fear index developed by Fuglenes, to measure the impact of perceived risk of complaints and malpractice litigation on providers' choice of MoB.[16]

Ten studies explored FoL's impact on MoB decision making beyond providers' self-report. The proxy measures used to explore FoL included changes to malpractice insurance premiums,[38 39 41–43] previous experience with litigation[19 22 44 45] and changes to liability laws.[22 46]

### Drivers of fear of litigation
Using the WHO principles,[25] we organised the drivers under three domains: (1) cognitive or psychological drivers; (2) sociocultural drivers and (3) environmental drivers (figure 2). For each of the three domains, table 2 shows the key drivers for FoL identified in this review, their definitions, and frequency and countries from which the findings derive. The number of citations supporting each driver is visualised in figure 3 showing that the legal environment (n=24), the patient pressure (n=19) and the availability bias (n=18) are the most frequently reported drivers in the literature. Findings for each domain are summarised below.

### Cognitive or psychological drivers
*Availability bias (in relation to experience with litigation)*
The impact of malpractice claims was discussed in 18 studies. In these accounts, personal experience of litigation was not essential for inciting FoL, rather 'hearing' about or knowing someone who was sued for malpractice could be enough.[18 36 44] Malpractice suits can be highly publicised events in the media and within the healthcare system[18 35] even though their impact is not always tied to adverse judicial outcomes for the provider. The mere thought of the experience of being sued due to failure or delay in performing CS, or the hypothesised impact on providers' time and reputation alone influenced decision making towards a CS.[47 48]

Some authors have quantified the impact of malpractice claims on defensive medicine practices among providers beyond personal experience showing mixed results. Brown found that one standard deviation increase in lawsuits was associated with a 1.073 increase in the risk of performing CS in USA.[19] Also in the USA, Cheng et al reported that clinicians who were sued were more likely to recommend CS than those who had not gone through this experience (17.2% vs 11.3% respectively, p=0.008)[38] while Dranove and Watanabe found a modest impact of litigation experience on CS rates of sued obstetricians and their immediate hospital colleagues.[44]

*Ambiguity aversion: CS 'just in case', avoid 'worst case scenario'*
Ambiguity, in the form of uncertainty of length and outcome of vaginal birth (VB), was a justification for preferring CS to avoid litigation consequences in 16 studies.[13 14 32 34 48–59] This tendency to avoid ambiguity is what leads professionals to perform CS, especially in the context of caesarean delivery on maternal request (CDMR)[58] or vaginal birth after CS (VBAC) or risk of uterine rupture.[34 57]

This ambiguity also may lead providers to shift the responsibility of decision making to the women and their families, rather than giving their professional opinion, as a means of avoiding litigation in case of complications; a CS becomes a sort of insurance policy.[58] Some authors note that this behaviour is reinforced by the fact that women and their families are increasingly perceiving adverse events in maternity care as unacceptable.[13 54 57]

*Relative risk bias and beliefs around safety*
Seven studies explicitly discussed providers' beliefs around the relative safety of CS.[37 41 51 54 60–62] The inclination to perform CS to avoid litigation risk can be motivated by providers' belief that CS is a safer option than VB, and that CS guarantees the best outcome for mother and child.[63] This belief can supersede training, guidelines, or statistical evidence demonstrating the opposite.[64]

In LMICs, this belief is further reaffirmed by the reduction in maternal morbidity and mortality due to the introduction—and framing—of CS as a lifesaving intervention in these countries.[60] Guidelines and research outlining the benefits of VB could be dismissed by providers in

**Table 2** Behavioural drivers of fear of litigation (FoL) under the three domains of the WHO principles, definitions and frequency and countries from which the findings derive

| Domain/construct | Definition | Frequency and citations | Countries |
|---|---|---|---|
| 1. Cognitive or psychological drivers: The identified constructs fall under heuristics, which as decision-making strategies rely on a few relevant predictors, while ignoring others. These shortcuts are helpful in some cases but may lead to cognitive biases, that is, a systematic error in judgement when processing information regarding the world around us that eases the cognitive burden of judgement and decision making.[24] | | | |
| 1.1. Availability bias (in relation to experience with litigation) | To make judgments of likelihood or frequency based on ease of recall rather than on actual probabilities and pertains to both direct experience, or vicarious through others.[24] | 18 ([14 16–19 22 35 36 38 42 44 45 47 48 63 80 88 89]) | Brazil, Iran, Israel, Norway, Sudan, Turkey, UK and USA |
| 1.2. Ambiguity aversion: CS 'just in case', avoid 'worst case scenario' | The tendency to prefer known or certain probabilities over uncertain probabilities regardless of the actual benefits.[24] | 16 ([13 14 32 34 48–59]) | Brazil, Canada, China, Iran, Kenya, Nicaragua, Paraguay, Sweden, Taiwan, Turkey, UK and USA |
| 1.3. Relative risk bias and beliefs around safety | Stronger inclination to (a birth mode) when presented with the relative risk than when presented with the same (information) described in terms of the absolute risk.[24] | 7 ([21 37 41 51 54 60 62]) | Argentina, Bangladesh, Brazil, Italy, Romania, Spain and Turkey |
| 1.4. Commission bias | Tendency towards action vs inaction. Omission bias: where harmful commissions are usually judged harsher than the corresponding omissions.[24] | 6 ([14 35 37 50 53 64]) | Brazil, India, Iran, Paraguay, Spain and Turkey |
| 1.5. Loss/gain framing or loss aversion bias | Losses are often perceived as looming larger than corresponding gains.[24] | 4 ([14 32 37 53]) | Brazil, Nicaragua, Paraguay and Spain |
| 2. Social and cultural drivers | | | |
| 2.1. Patient pressure | Pregnant women, or their families exerting pressure on providers to perform Caesarean delivery on Maternal Request (CDMR), for non-medical reasons. | 19 ([32 47 53 56 58–63 66–74]) | Bangladesh, Brazil, Canada, France, Germany, India, Iran, Italy, Luxembourg, Netherlands, Nicaragua, Paraguay, Romania, Spain, Sweden, Taiwan, UK and USA |
| 2.2. Social norms: CS as the new 'normal birth' | Implicit or explicit rules used by a group and which determine values, beliefs, attitudes and behaviours.[90] These norms can be descriptive, that is, what people observe as the typical overt behaviour of others, or injunctive norms, that is, inference of what is expected of an individual by others.[91] | 8 ([14 32 37 47 49 51 66 69]) | Brazil, China, Germany, Iran, Ireland, Italy, Nicaragua and Spain |
| 2.3. Blame culture | A culture that does not tolerate error can be a driver to medical excesses, and negative impacts on healthcare quality and patient safety.[92 93] | 10 ([17 32 34 50 52 57 58 69 71 75]) | Iran, Ireland, Italy, Netherland, Nicaragua. Sudan, UK, USA and Sweden (mention of absence of blame culture) |
| 3. Environmental drivers: any factor external to the individual that could impact the behaviour. This could include physical environment, or social environment (eg, medical administrators, peers, patients and their families) | | | |
| 3.1. Legal environment | The legal framework within which providers practice and are faced with liability. This could include tort laws, court systems, admissible evidence and expert witnesses. This also covers legal mediation, negotiation and settlements outside the court systems. | 24 ([13 14 21 22 33–38 41 43 46 47 49 50 52 54 62 63 76–79]) | Brazil, China, Iran, Italy, Romania, Spain, Sweden, Turkey, UK and USA |

Continued

**Table 2** Continued

| Domain/ construct | Definition | Frequency and citations | Countries |
|---|---|---|---|
| 3.2. Malpractice insurance premiums | The existing insurance mechanisms against malpractice including degree of coverage and premiums. | 9 ([35 38 39 41 43 54 56 79 80]) | Argentina, Canada, Turkey, USA |
| 3.3. Medical and professional environment | The medical system which organises obstetric practices through medical guidelines, obstetric organisations or health facility administration. | 7 ([18 32 34 37 44 52 80]) | Iran, Israel, Nicaragua, Spain and USA |
| 3.4. Mass and social media | This includes social and traditional mass media. | 5 ([14 18 35 63 66]) | Brazil, Israel, Turkey and USA |

LMICs as perceived not reflecting the realities of their experiences.[50]

Even when providers acknowledge the comparative disadvantages of CS to mother and child in the long term, they may not share this information with their clients.[50] As a consequence, the perceived short-term disadvantages of VB are exaggerated or overstated, making VB the less desirable option for providers in light of the risk of litigation, even when VB is objectively safer for women and babies.[32]

*Commission bias*

Six studies identified drivers that described a tendency to perform CS even if medically unnecessary, just to 'do something' rather than wait for uncertain labour events or outcomes.[14 35 37 50 53 64] With rising rates of CS and lower tolerance for negative maternity care outcomes, providers find waiting for VB challenging, especially if labour is not progressing as expected.[50]

Such findings can be categorised as commission bias in maternity care despite evidence that performing CS without medical indication does not reduce the risk of lawsuits.[65] It contrasts with 'omission bias' in other medical specialties,[24] where harmful commissions are usually judged harsher than the corresponding omissions. In case of complications, maternity care providers perceived that they are more likely to be sued if they failed to perform a CS or delayed it, but unlikely to be sued for performing an unnecessary CS. A study surveying 403 obstetricians in Brazil reported that 67.5% believed VB poses a higher risk of litigation in case of complications, while only 0.5% found that CS had a higher litigation risk.[14] A Turkish study found that the perception of VB as a litigation risk was reinforced by wide media coverage of legal cases focusing on delay or failure to perform a CS.[35]

*Loss/gain framing or loss aversion bias*

Four studies discussed the substantial direct and indirect costs associated with litigation, regardless of the litigation outcome, which underpinned the loss aversion bias pervasiveness in maternity care. Losing a case might result in losing medical licensure, and/or paying large settlements.[14 35 47] Moreover, regardless of the legal outcome, providers might have to pay higher premiums for malpractice insurance,[39] face negative impacts on their reputation[42 44] and suffer the time and psychological burden of the litigation process itself.[38 47]

### Sociocultural drivers
*Patient pressure*

Nineteen studies explicitly examined FoL as a driver for providers to agree to a CS based on maternal request and/or pressure by mothers and families.[32 47 53 56 58–63 66–74]

Providers' attitudes towards CDMR varied between some providers believing women have a right to choose how they give birth, to others asserting that medical necessity supersedes service users' preferences.[62 73] A study conducted in Italy aimed to explore the perceptions of provider legal liability in case of CDMR among obstetricians, midwives, lawyers and patients.[72] On average, patients and lawyers rated concession to CDMR as highly appropriate, in contrast to providers who found it inappropriate. Patients and lawyers also rated judicial responsibility of the providers for complications after a CDMR lower than after a VB, suggesting that the provider's

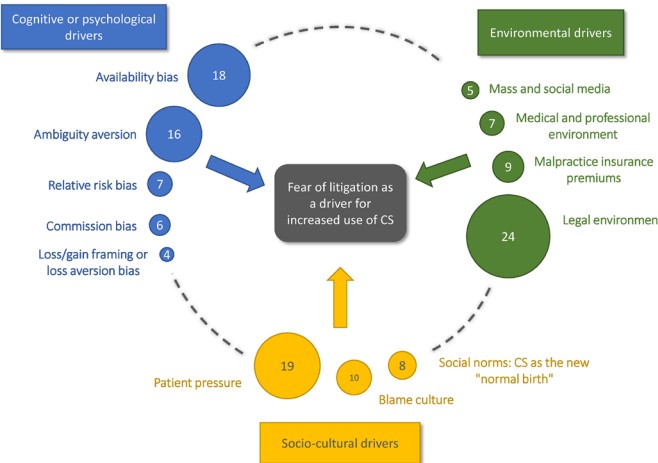

**Figure 3** Number of citations supporting each driver among 56 citations included in this scoping review. CS, caesarean section.

compliance with maternal choice can be the deciding factor in the litigation process. Providers reported that respecting maternal wish for CS is more likely to help avoid litigation in case of complications.

Women's socioeconomic status and education also impacted the providers' perception of their likelihood to be sued in case of complications. Brazilian providers described how more educated women have a higher expectation of care and intolerance to complications during birth, both of which lead to increased risk of litigation if their demands for a CS are not met.[66]

### Social norms: CS as the new 'normal birth'

Eight studies reported providers' concerns that CS is becoming a social norm among women and relatives, since it is widely regarded as safer and 'more modern' than VB.[14 32 37 47 49 51 66 69] In two studies in China and Brazil, providers claimed that CS is expected by service users and families, posing substantial pressure on providers.[14 49] It has been observed that, as countries become more economically stable, families have less children and acceptance or tolerance of adverse events during labour is reduced.[18 47 49] In these contexts where CS may be seen as the norm, the perception of VB by providers as riskier increases its perceived litigation risk and decreases a provider's ability to defend their performance following a complication.[32 66]

### Blame culture

Six studies explicitly addressed the effect of blame culture on FoL in relation to CS choice,[17 32 50 52 69 71] in addition to four studies implicitly discussing the effects of blame culture.[34 57 58 75] This culture can result in a hesitancy to even suggest VB as an option for fear of blame in case of complications.[58]

In countries where providers expressed less fear of being blamed, they also reported less inclination to perform a CS. In a study on CS decision making in Sweden, midwives expressed that FoL is not a concern because the medical and legal environment does not foster placing blame, which improves transparency and empowers midwife-led care. Similarly, a study in multiple European countries reported that participants attributed the low rates of VBAC in their countries (Ireland and Italy) to blame culture, in comparison to higher rates in Sweden.[69] A qualitative study of three countries with high VBAC rates (Finland, Sweden and the Netherlands) showed that providers in the Netherlands were concerned with their decline in VBAC rates due to increased risk of litigation and blame in case of complications.[75]

### Environmental drivers

The third principle encompasses the external factors that enable or block a behaviour, including the legal environment, organisational culture, medical bodies policies, guidance, media and other larger environmental or structural factors that might influence the perceptions of the providers and their decisions (table 2).

### The legal environment

The legal environment and how it influences maternity care were the focus of 24 studies.[13 14 21 22 33–38 41 43 46 47 49 50 52 54 62 63 76–79] While laws, legal norms and procedures differ by context, we identified common themes across different studies.

The first theme depicted the perceived randomness of legal processes due to the absence of specialised courts, or of a clear legal framework for medical complaints,[14 47] or in some cases even a malpractice law.[14 34 35] These gaps resulted in a perception of lack of fair and specialised legal processes and reinforced providers' perceptions of their vulnerability in case of adverse event, contributing to FoL.

In addition, reliance on expert witnesses in court to testify regarding fault or negligence is a driver for FoL because of the perceived lack of rigour and criteria for assessing medical negligence or malpractice, or understanding the variability in medical opinions.[16] In a study in Iran, providers expressed discontent with the lack of training or knowledge of expert witnesses whom they perceived as chosen based on personal relationships, and as having no interest in examining the causes of adverse events, but who rather relied on emotional criteria.[47]

The second theme illustrates the lack of clear demarcation within legal systems between malpractice and unavoidable adverse events.[54] Providers in some of the included studies perceived that judges, jurors and expert witnesses assign fault based on empathy with the families' suffering a loss or disability following birth, rather than looking at whether or not this negative outcome was a result of negligence.[34] This perception could be reinforced in contexts where limited social or financial support to families and children with disabilities existed, and hence the legal environment seeks to place blame as a way to help the family with the resulting settlement.[12]

Once again, this legal environment results in providers considering CS a safer option. If sued after a VB, providers believe they will need to justify not having conducted a CS and whether a CS could have prevented the negative outcome.[50 72] On the other hand, performing a CS when not medically necessary is not recognised as malpractice.[16 50 72] In addition, depending on the legal context, providers can be at risk of liability for fault even years after the birth. This adds to providers' long-term perceived risk of not performing a CS.[47]

As more cases are settled in favour of families based on adverse outcomes, and more service users are aware of their right to sue if the birth does not go according to their expectations, the legal and financial incentives for litigation are mounting.[38] In five studies, providers claimed that the settlements themselves have become an incentive for more litigation, and for providers to avoid VB for FoL.[14 37 42 47 49]

### Malpractice insurance premiums

Malpractice insurance premiums and their relationship to FoL was discussed in nine studies.[35 38 39 41 43 54 56 79 80] A study in the USA found that the likelihood of having a CS

increased, and the likelihood for a VB or an instrumental VB decreased, in states where insurance premiums for malpractice exceeded 100 000 US$ compared with states where insurance premiums were below 50 000 US$. Inversely, another study found that higher malpractice insurance premiums were not positively associated with recommending CS, but rather that providers who did not know how much they paid for malpractice insurance premiums were less likely to recommend CS, suggesting a possible lesser preoccupation with malpractice risk.[38]

### Medical and professional environment

Seven studies discussed medical and professional environments in which providers reported lack of or little protection afforded to them by medical bodies and authorities.[18 32 34 37 44 52 80] Medical providers report feeling discouraged or 'demoralised' for having to face these risks without the backing of their colleagues and scientific authorities for their clinical decisions in societies that are becoming increasingly litigious.[37] Three articles also mention that absent or conflicting guidelines regarding CS decision making contributed to their FoL.[32 47 80] However, administrators in a study in Nicaragua argued that the guidelines exist but are not followed.[32]

### Mass and social media

Media was mentioned as an influential actor in motivating FoL associated with MoB in five studies.[14 18 35 63 66] The sensationalising of malpractice claims incited fear and frustration among providers. A study in Turkey found that all included providers attested to practicing defensive medicine and being affected by recent high profile malpractice cases in media which resulted in large settlements.[35] This media effect, whether traditional or social media, went beyond fearing the risk of litigation, to having a serious impact on providers' mental health, reputation and their approach to their profession.[14]

## DISCUSSION

This scoping review gathered the evidence available on drivers of FoL when deciding for MoB from the perspective of behavioural theory. The findings highlight the complexity of FoL as a construct that contributes to rising rates of CS across the world. Our use of behavioural principles to develop a conceptual framework for examining drivers at different levels, aided in understanding how they can interact to influence providers' behaviours and choices (figure 2). This behavioural lens also lent itself to identifying cognitive influences and biases that, while not always explicitly mentioned in the literature, were interpreted through providers' accounts. Identifying such bias could inform the design of behavioural interventions aimed at influencing the driver of FoL.

This review exposed the underlying factors—and the domains they pertain to—which build FoL motivating providers' decision to perform a CS beyond medical need. While our focus was litigation, it is hard to extricate these fears from perceptions of relative safety of CS, fears of disrepute and consequences of the process of litigation itself rather than its outcome. This more nuanced understanding of fear as a driver for CS is crucial to address unnecessary surgical births globally.

The review exposed the need for a consensus around measuring FoL in maternity care, given the diversity of proxies used such as malpractice premiums or experience of litigation. Most studies relied on providers self-reported subjective definition, understanding and perception of FoL, without a standardised scale or comparison of actual litigation risk and outcomes. Relying on self-report poses a challenge to reliably quantify and compare the magnitude or burden of FoL, particularly given the multitude of other factors driving the increasing CS rates and which may be beyond the providers' control.[81 82] Using more complex measures such as a 'fear index' as composite measures of FoL as proposed by Fuglenes et al[16] could provide some value if used in conjunction with other objective measures of litigation risk such as malpractice claims.

Healthcare providers have to rapidly integrate and interpret a complex range of social, psychological, clinical and emotional data relating to both the person they are caring for and the wider context in which they are working, in continuously changing situations where often little time is available to make decisions. In maternity care, this is intensified by the fact that the care provider is undertaking this task for both mother and fetus during both pregnancy, labour and birth, and balancing the sometimes-differing needs of both. Despite existing guidelines, fast decisions require intuitive thinking processes, that jump to the most likely interpretation of and solution to a problem based on patterns of expert knowledge rather than following sequential rules of rationality.[83] This review suggests that, in terms of decision for MoB, emotional responses and primarily fear can be driven by cognitive biases including rational and irrational beliefs regarding safety and risk. These beliefs can supersede training, guidelines or statistical evidence and are deeply entrenched in sociocultural structures.[12 24] Crucially, the review confirms the influence of sociocultural and environmental factors which encompasses multiple stakeholders including women, families, health professionals, healthcare organisations and legal systems. Hence multilevel approaches beyond knowledge-based intervention or training are required to address FoL.[84]

This scoping review unveiled the diversity of drivers behind fear as experienced by providers, the domains to which these drivers belong as, the actual risk of litigation and how it manifests in medical practice. Further contextual studies are warranted for in-depth understanding of local dynamics. However, the range of contexts included in the review suggests that underlying concepts revealed in the data synthesis are potentially transferable between quite different contexts.

Our review also highlights the close relationship and interaction between FoL and shifts in norms, particularly

through the role and power of media either mainstream or social. Our findings suggest that trial-by-media may be regarded as having a very real immediate influence while the consequences of any litigation process may take years and are ambiguous. The increasing tendency for sensationalist media coverage, and 'click-bait' reinforces this trend.[85] Media may also feed into the public perception that CS is the safest and most modern option, and that women should have a choice of elective CS, while downplaying the need to inform people about the risks associated with this choice (considering it 'fear mongering'). Media pressures may also act to divorce the sense of entitlement to a right to choose from the personal responsibility to accept the consequences for such choices.[86] This means that any adverse outcome, even if resulting from personal choices, is deemed to be the fault of someone else, and thus ripe for a litigation case. Further research into how these fears and factors operate and reinforce each other is needed.

Our focus on behavioural drivers for providers means we did not include perspectives of service users and their families, which may be quite different. Although accounts of providers are necessary to understand drivers of FoL, they do not represent an objective measure of the strength or frequency of the phenomena described. For example, while 20 of the included studies discuss CDMR as an important driver of increasing CS rates, global rates of CDMR show great variability across regions (0.2%–42%), with upper middle income countries having 11 times the rate of CDMR compared with HICs.[87]

### Strengths and limitations

To our knowledge, this is the first review to systematically explore global accounts of FoL and its behavioural drivers in detail, providing a broad overview of the evidence on this multidisciplinary topic and identifying research gaps. The use of the WHO behavioural principles enabled us to classify behavioural drivers within specific domains offering clearer pathways for the design of interventions and policies.

Our review has some limitations. We might have overlooked evidence from grey literature and other informal publications. However, we found and presented data from a range of high-income, middle-income and low-income countries. Our classification of cognitive influences on FoL relied on commonly used behavioural constructs and terminologies in medical decision-making literature.[12 24] These constructs may have not been explicitly used by the authors of the studies included to classify their findings, or by participants to describe their own rationale for MoB decision making. In order to avoid forcing false categories on the data, we were very conservative in how we classified findings and explicitly sought data that disconfirmed prior assumptions. This was facilitated by the fact that a few distinct themes resonated in the articles across settings. On the other hand, local sociocultural and environmental influences were often explicitly mentioned in the reviewed articles, and in some cases, were the focus of the study. Although our synthesis showed consistent themes across settings, any initiatives or strategies to address FoL need to be tailored to the specific local context to address the particularities of legislative and health systems and consider how behaviour change interventions can be operationalised locally. Lastly, while we were interested in understanding the drivers of medically unnecessary CS, the reviewed literature often did not differentiate between necessary and unnecessary CS, or document the rates of unnecessary CS. Hence, we have approached our study of drivers of FoL in relation to 'increasing use of CS' rather than 'increasing rates of unnecessary CS'.

## CONCLUSION

FoL as a driver for rising CS rates is the result of a complex interaction between cognitive, social and environment factors. Factors identified across countries include cognitive bias, social pressure from service users and prevailing norms such as growing intolerance to complications and uncertainty, legal and medical practice environments, and experience with litigation. FoL may be generated less by the actual risk of litigation and more by how the providers perceive this risk and its potential consequences of the process itself independently of the actual legal outcome. Behavioural interventions addressing these drivers are likely to be crucial to address FoL as part of strategies aimed at reducing CS rates.

**Author affiliations**
[1]Community Medicine Department, Alexandria University Faculty of Medicine, Alexandria, Egypt
[2]Behavioural Insights Unit, World Health Organization, Geneve, Switzerland
[3]School of Community Health & Midwifery, University of Central Lancashire, Preston, UK
[4]Schulich School of Law, Dalhousie University, Halifax, Nova Scotia, Canada
[5]School of Medical Sciences, University of Hyderabad, Hyderabad, India
[6]Department of Medicine, Sao Paulo Federal University, Sao Paulo, Brazil
[7]UNDP/UNFPA/UNICEF/World Bank Special Program of Research, Development and Research Training in Human Reproduction (HRP), Department of Sexual and Reproductive Health and Research, World Health Organization, Geneva, Switzerland

**Acknowledgements** With thanks to Jane Gibbon, Sarah Cordey and Louise Hunt.

**Contributors** All authors included on a paper fulfill the criteria of authorship. SE, EA, SD and APB conceived and plan the review. SE and APB screened the studies. SE conducted data extraction and analyses. SE wrote the first draft of the manuscript with support from EA and APB. SD, JE, SM, GM, BRS, MRT made substantial contributions to interpretation, and critically reviewed the manuscript for important intellectual content. SE acts as guarantor.

**Funding** This scoping review was funded by the Behavioural Insights Unit at World Health Organization, the UNDP-UNFPA-UNICEF-WHO-World Bank Special Programme of Research, Development and Research Training in Human Reproduction (HRP), a co-sponsored program executed by the World Health Organization (WHO) in the Department of Sexual and Reproductive Health and Research (SRH) and the Gates Grand Challenges 2020: Improving Access to and Use of Safe and Appropriate Cesarean Section (INV-023309). The review was conceived to inform the project Re-JUDGE (https://gcgh.grandchallenges.org/grant/re-judge-reducing-rates-non-medically-indicated-cesarean-sections-through-open-access-multi).

**Competing interests** None declared.

**Patient and public involvement** Patients and/or the public were not involved in the design, or conduct, or reporting, or dissemination plans of this research.

**Patient consent for publication** Not applicable.

**Ethics approval** Not applicable.

**Provenance and peer review** Not commissioned; externally peer reviewed.

**Data availability statement** All data relevant to the study are included in the article or uploaded as supplementary information.

**ORCID iDs**
Elena Altieri http://orcid.org/0000-0001-6603-3399
Maria Regina Torloni http://orcid.org/0000-0003-4944-0720
Ana Pilar Betran http://orcid.org/0000-0002-5631-5883

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
