## [Reviewer comments · BMJ Open]

ARTICLE DETAILS

TITLE (PROVISIONAL)	Behavioural factors associated with fear of litigation as a driver for the increased use of caesarean sections: A scoping review
AUTHORS	Elaraby, Sarah; Altieri, Elena; Downe, Soo; Erdman, Joanna; Mannava, Sunny; Moncrieff, Gill; BR, Shamanna; Torloni, Maria Regina; Betran, Ana Pilar

VERSION 1 – REVIEW

REVIEWER	Rasool, Muhammad Bahauddin Zakariya University
REVIEW RETURNED	10-Jan-2023

GENERAL COMMENTS	The article "Behavioural factors associated with fear of litigation as a driver for the increased use of caesarean sections: A scoping review" addresses a very important topic. The review is very well written and is worth publishing. I have following comments, 1. The method section does not state how the quality of the included studies was checked?2. How was the selection Bias addressed?3. Figure 1 kindly use uniform text style.4. Figure 2, kindly use different colors in the background or use different color for the text as it is hard to follow.
---

REVIEWER	Khowaja, Bakhtawar The Aga Khan University, Obstetrics and Gynecology
REVIEW RETURNED	10-Jan-2023

GENERAL COMMENTS	I would like to thank the researchers for spending time on this important subject. It is frustrating, but important, to read how many factors including FoL as one of those, are to influencing the high CS rates, an area where a lot of work is needed. This review gives insights and is relevant for strengthening and developing future work on legal and audit frameworks, especially in maternity specialty. However, revisions are necessary to convince the readers. My comments are provided below which can help researchers to improve the readability of this important work Introduction Line 5: Could you justify the increase in the section rates globally by using some numbers and statistics? This is quite interesting and surprising to know that obstetrics is the leading specialty in terms of litigation. Could you please provide few examples to interest readers
--

	Please elaborate more on the research team and their specific work on this review. Two independent researchers reviewed for discrepancies, what about the rest of pieces such as development of themes and etc. The tabular demonstration represents that there were 2 studies from the administrators' perspectives, but I didn't find anything in the results section from their viewpoint. Was it same as of other healthcare providers or it wasn't worth mentioning? How did you develop the themes? Please provide a thematic map on how you developed the themes from the WHO principles, and I am also interested to see that how you merged 6 principles into 3. Elaborate a bit on methodology part Were there any cultural factors especially from LMICs along with the social factors? I would prefer changing objective to increasing CS rates rather than influencing CS rates because the FoL is the cause of only rising trends What is the custom form mentioned in the abstract? Please define/elaborate terms such defensive medicine, omission and commission bias. Some readers might not understand their meaning I dont understand figure 2. Would rather prefer having a thematic map instead Inclusion criteria: include peer-reviewed articles, (thesis/dissertation and grey literature for India review only), why India only? Why have you excluded case studies/case series? I am interested to know if you were able to see any difference of findings in low and high income countries. I am sure there must be some which you could state in results as well as discussion section. Conclusion: Please include a short answer to the research question. Good luck with the revisions and do not hesitate to contact me if there are questions.
--	--

REVIEWER	Grassi, Simone Università Cattolica del Sacro Cuore, Department of Health Surveillance and Bioethics
REVIEW RETURNED	07-Feb-2023

GENERAL COMMENTS	I did appreciate the review because it is innovative and appealing for hospitals' decision-makers. I suggest to the authors to make minor amendments to enhance its methodological rigour. Regarding the inclusion criteria, the authors should specify why they also considered papers written in Spanish, Portuguese and French. At page 37 the authors indicate as inclusion criterion German as publication language but this criterion has not been
---

	reported before. The authors should explain this fact. The authors should also be transparent about the choice of including thesis/dissertation and grey literature for India review only. Moreover, the authors should specify why they included only papers reporting more than 6 cases. It should be specified graphically how many papers supported each of the flagged drivers.
--	---

VERSION 1 – AUTHOR RESPONSE

Reviewer: 1 (Dr. Muhammad Rasool, Bahauddin Zakariya University) The article "Behavioural factors associated with fear of litigation as a driver for the increased use of caesarean sections: A scoping review" addresses a very important topic. The review is very well written and is worth publishing. I have following comments, 1. The method section does not state how the quality of the included studies was checked? 2. How was the selection Bias addressed?	Thank you for your considerations and positive feedback. This is a scoping review. The general purpose for conducting scoping reviews is to provide a broad overview on a given topic, and give an indication of the volume, nature or diversity of available literature and its focus. They tend to be a precursor to a systematic review. Unlike a systematic review, scoping reviews do not tend to produce and report results that have been synthesized following a formal process of methodological appraisal to determine the quality of the evidence (see reference below). Due to this, and according to international guidelines, an assessment of methodological limitations or risk of bias of the evidence included within a scoping review is generally not performed. Subsequently, the implications/recommendations for practice (from a clinical or policy making point of view) that arise from a scoping review are quite different compared to those of a systematic review. We have added in the methods section that we followed the standard methodology for scoping reviews as follows: “The scoping review followed the standard methodology recommended by the Joanna Briggs Institute (see reference below) and is reported in adherence to Preferred Reporting Items for Systematic Reviews and Meta-Analyses for Scoping reviews (PRISMA-ScR)” REFERENCE: Peters MDJ, Godfrey C, Mclnerney P, Munn Z, Tricco AC, Khalil, H. Chapter 11: Scoping Reviews (2020 version). In: Aromataris E, Munn Z (Editors). JBI Manual for Evidence Synthesis, JBI, 2020. Available from https://synthesismanual.jbi.global. https://doi.org/10.46658/JBIMES-20-12
3. Figure 1 kindly use uniform text style.	As requested, the font in Fig 1 has been homogenized (the same font has been used throughout the boxes in the figure).
4. Figure 2, kindly use different colors in the background or use different color for the text as it is hard to follow.	As requested by the reviewer, Fig 2 has been redesigned (particularly the colors) for improved reading.
Reviewer: 2 (Ms. Bakhtawar Khowaja, The Aga Khan University)	Thank you for your considerations and positive feedback on the importance of this topic and the need to address it. As requested by the reviewer, we have added in the introduction the following sentence to show the global increase in the last decades (and the corresponding reference):

I would like to thank the researchers for spending time on this important subject. It is frustrating, but important, to read how many factors including FoL as one of those, are to influencing the high CS rates, an area where a lot of work is needed. This review gives insights and is relevant for strengthening and developing future work on legal and audit frameworks, especially in maternity specialty. However, revisions are necessary to convince the readers. My comments are provided below which can help researchers to improve the readability of this important work. Introduction Line 5: Could you justify the increase in the section rates globally by using some numbers and statistics?	“Latest available estimates show that worldwide, the average CS rate increased from 6.7% in 1990 to 21% in 2018” REFERENCE: Betran AP, Ye J, Moller A-B, et al. Trends and projections of caesarean section rates: global and regional estimates. BMJ Global Health 2021;6:e005671. doi:10.1136/ bmjgh-2021-005671
This is quite interesting and surprising to know that obstetrics is the leading specialty in terms of litigation. Could you please provide few examples to interest readers	As requested by the reviewer, we have added additional references (see below) showing examples for the interested readers.  • Montilla P, Merzagora F, Scolaro E, et al. Lessons from a multidisciplinary partnership involving women parliamentarians to address the overuse of caesarean section in Italy. BMJ Global Health 2020;5:e002025. • Lane, J., Bhome, R., & Somani, B. (2021). National trends and cost of litigation in UK National Health Service (NHS): a specialty-specific analysis from the past decade. Scottish Medical Journal, 66(4), 168-174. • Almannie R, Almuhaideb M, Alyami F, Alkhayyal A, Binsaleh S. The status of medical malpractice litigations in Saudi Arabia: Analysis of the annual report. Saudi J Anaesth. 2021 Apr-Jun;15(2):97-100. doi: 10.4103/sja.SJA_908_20. Epub 2021 Apr 1. PMID: 34188624; PMCID: PMC8191242.
Please elaborate more on the research	As requested by the reviewer, we have detailed in the methods section who in the team was involved in each step conducted:

team and their specific work on this review. Two independent researchers reviewed for discrepancies, what about the rest of pieces such as development of themes and etc.	“Titles and abstracts were screened in duplicate, and full texts of potentially relevant studies were obtained and assessed by two reviewers independently (SE and APB). Discrepancies were discussed until consensus was reached.” “One reviewer (SE) supervised by two reviewers (EA and APB) conducted the data extraction and developed the codebook”. “[Data synthesis] This process was conducted by one reviewer (SE) and discussions among co-authors informed the classification of drivers according to the WHO principles and the definition of how they interact” The specific work of the research team in this review is detailed in the contributorship statement as follows (names initials have been used for abbreviation): “SE, EA, SD and APB conceived and plan the review. SE and APB screened the studies. SE conducted data extraction and analyses, SE wrote the first draft of the manuscript with support from EA and APB. SD, JE, SM, BRS, MRT made substantial contributions to interpretation, and critically reviewed the manuscript for important intellectual content”.
The tabular demonstration represents that there were 2 studies from the administrators' perspectives, but I didn't find anything in the results section from their viewpoint. Was it same as of other healthcare providers or it wasn't worth mentioning?	Thanks for this valuable remark. While two qualitative studies specified administrators as part of their study population, they did not emphasize the differences in insights shared by obstetrician/gynecologist (clinicians) vs. administrators. See the citations of the two studies below as well as a brief description of the two studies. In the study by Munro et al., a majority of administrators/decision-makers had clinical experience providing obstetric care, hence they provided insights reflecting both their clinical practice and their decision-making responsibility. The article’s only mention of variation in accounts stated that: “Care providers’ interviews revealed the characteristics of clinical decision making while decision makers highlighted the influence of health service resources and policy on the context of that decision making.” We assessed this was insufficient to substantiate results of the review. The study by Colomar et al. only highlighted one variation in opinion as well: “Practicing physicians felt that there was a lack of clear clinical protocols/guidelines on the indications for cesarean birth, particularly for a woman who has had a previous cesarean. Decision-makers indicated that clinical guidelines do exist, so either the guidelines have not been widely disseminated or obstetrician/gynecologists simply do not feel adequately supported to apply them.” However, in view of the comment by the reviewer, we have added a sentence recognizing these views under the Environmental drivers section (Medical and professional environment) as below: “Three articles also included mention that absent or conflicting guidelines regarding CS decision making contributed to their FoL (38, 43, 60). However, administrators in a study in Nicaragua argued that the guidelines exist but are not followed (38).” REFERENCES: Munro S, Kornelsen J, Corbett K, Wilcox E, Bansback N, Janssen P. Do Women Have a Choice? Care Providers' and Decision Makers' Perspectives on Barriers to Access of Health Services for Birth after a Previous Cesarean. Birth (Berkeley, Calif). 2017;44(2):153-60.

Colomar, M., Cafferata, M.L., Aleman, A. et al. Mode of Childbirth in Low-Risk Pregnancies: Nicaraguan Physicians' Viewpoints. *Matern Child Health J* 18, 2382–2392 (2014). <https://doi.org/10.1007/s10995-014-1478-z>

How did you develop the themes? Please provide a thematic map on how you developed the themes from the WHO principles, and I am also interested to see that how you merged 6 principles into 3. Elaborate a bit on methodology part

We did not merge WHO principles, rather we focused on the first three. Principles 1-3 explain factors that underpin, influence, and enable or hinder human behaviour, and hence were a suitable framework for our article's aims. On the other hand, principles 4-6 focus on harnessing the behavioural knowledge on human behaviour to design and implement interventions, which is outside the scope of our work. See below the 6 principles of behavioural insight for health and the reference.

The three principles acted as parent themes under which we developed deductive subthemes in the initial codebook. During the analysis, we added additional inductive themes and revised prior deductive ones. Through an iterative process, which included continuous discussions within our team, we pared down the resulting subthemes by cutting themes that were outside the scope of the review, merging ones that were redundant, and excluding themes that were supported by data from one study only.

We edited the methods section to include these details to the analysis.

6 PRINCIPLES OF BEHAVIOURAL INSIGHT FOR HEALTH

Ref: Technical Note from the WHO Technical Advisory Group (TAG) on behavioural insights and science for health. Principles and steps for applying a behavioural perspective to public health: World Health Organization; 2021 [Available from: https://cdn.who.int/media/docs/default-source/documents/bi-tag-technical-note1_principles-and-steps.pdf?sfvrsn=efdefb39_5&download=true.]

	We followed a very iterative process including the various experts of the team. As expected, many versions were produced which can be shared but we don't think it would be appropriate to publish.
Were there any cultural factors especially from LMICs along with the social factors?	We did not seek to delineate the difference between cultural and social factors, both of which fall under one principle of the WHO framework (Social and cultural context – See the figure above under the response to the previous comment). There were some examples of cultural factors that we included such as blame culture within the medical system, which is prevalent in many LMICs. Additionally, in some articles cultural factors motivated maternal demand for a CS, such as rising concerns about the impact of vaginal delivery on sexual life or desiring to deliver on a specific date to improve the child's fortune in life. We chose not to discuss these factors in details, since they were not specifically associated with fear of litigation, but rather underpinned other influences already discussed.
I would prefer changing objective to increasing CS rates rather than influencing CS rates because the FoL is the cause of only rising trends	We prefer to use the word "influencing" as it is more inclusive. "Increasing" limits the direction of the changes. In the current context of very high CS rates in many countries (e.g. national rates in Brazil, Cyprus, the Dominican Republic, Turkey, Iran, Chile, Lebanon, or Paraguay have surpassed or are around 50%), factors may influence not only the increase but also the lack of reduction in the CS rates when implementing interventions aiming at reducing rates.
What is the custom form mentioned in the abstract?	We apologized for the lack of clarity. We have rephrased the abstract as follows: “Data extraction and synthesis: Data were extracted using a form specifically designed for this review and we conducted content analysis using textual coding for relevant themes.”
Please define/elaborate terms such defensive medicine, omission bias and commission bias. Some readers might not understand their meaning	Thank you for this comment. We understand some readers may be unfamiliar with this terminology. You will find the definition of commission bias in Table 2. We have added the definition of omission bias to this table. We have added a definition of defensive medicine in the Introduction (3rd paragraph) where the term is first mentioned. We have defined the terms:  • Commission bias: In Table 2: “When presented with an unclear decision, medical providers could have a higher tendency towards action versus inaction” • Omission bias: In Table 2: “where harmful commissions are usually judged harsher than the corresponding omissions” • Defensive medicine: In introduction: “practice wherein a healthcare professional makes decisions out of fear of litigation and not for the benefit of the patients”
I dont understand figure 2. Would rather prefer having a thematic map instead	Figure 2 summarizes the three different types of behavioural factors as identified in the review and detailed in Table 2. These three types of factors influence fear of litigation which in turn influences the increase use of caesarean section. Similar frameworks are used in the literature (e.g. COM-B model) and we would suggest to keep it since it is consistent with the WHO 3 internal principles of behavioural insight for health outlined in the figure above in previous response.

Inclusion criteria: include peer-reviewed articles, (thesis/dissertation and grey literature for India review only), why India only?	Thank you for noting this typo. This is incorrect and an error on our side. We have deleted it. We did not include grey literature or dissertations in this scoping review.
Why have you excluded case studies/case series?	Thank you for the question. We decided to exclude case studies/case reports because this type of literature provides very limited information and other types of study design would be more useful and appropriate for this review topic. According to the JBI manual (Chapter 11, section 11.2.4 Selection Criteria), review authors are free to impose limits on the types of studies included in scoping reviews. REFERENCE: Peters MDJ, Godfrey C, McInerney P, Munn Z, Tricco AC, Khalil, H. Chapter 11: Scoping Reviews (2020 version). In: Aromataris E, Munn Z (Editors). JBI Manual for Evidence Synthesis, JBI, 2020. Available from https://synthesismanual.jbi.global. https://doi.org/10.46658/JBIMES-20-12
I am interested to know if you were able to see any difference of findings in low and high income countries. I am sure there must be some which you could state in results as well as discussion section.	Unfortunately, a scoping review is not designed to measure or quantify these types of differences in a scientifically appropriate manner. However, Table 2 depicts the number of studies found under each behavioural factor and the names of the countries from which the studies come. From this table the reader can easily identify that no studies come from low-income countries. Most of them are from Europe, US and Latin America (the region with the highest CS rates).
Conclusion: Please include a short answer to the research question	As suggested by the reviewer, we have shortened and edited the Conclusion as follows: “FoL as a driver for rising CS rates is the result of a complex interaction between cognitive, social, and environment factors. Factors identified across countries include cognitive bias, social pressure from service users, and prevailing norms such as growing intolerance to complications and uncertainty, legal and medical practice environments, and experience with litigation. FoL may be generated less by the actual risk of litigation and more by how the providers perceive this risk and its potential consequences of the process itself independently of the actual legal outcome. Behavioural interventions addressing these drivers are likely to be crucial to address FoL as part of strategies aimed at reducing CS rates.” We have also edited the conclusion in the abstract.
Reviewer: 3 (Dr. Simone Grassi, Università Cattolica del Sacro Cuore) I did appreciate the review because it is innovative and appealing for hospitals' decision-makers. I suggest to the authors to make	Thank you for your considerations and positive feedback. Regarding the languages, we apologize for the lack of clarity and inconsistency. The intention was to include articles in German language but we did not find any article in German to include. This is the reason why we did not put it in the methods section of the manuscript; because no article in German was included. We have clarify this in the Supplementary material.

minor amendments to enhance its methodological rigour. Regarding the inclusion criteria, the authors should specify why they also considered papers written in Spanish, Portuguese and French. At page 37 the authors indicate as inclusion criterion German as publication language but this criterion has not been reported before. The authors should explain this fact. The authors should also be transparent about the choice of including thesis/dissertation and grey literature for India review only.	
Moreover, the authors should specify why they included only papers reporting more than 6 cases.	Please see the response to reviewer 2 for the inclusion criteria. Smaller sample sizes would typically fall under case studies or case series, which we excluded in this review. REFERENCE: Esene, I. N., Kotb, A., & ElHusseiny, H. (2014). Five is the maximum sample size for case reports: statistical justification, epidemiologic rationale, and clinical importance. World Neurosurg, 82(5), e659-65.
It should be specified graphically how many papers supported each of the flagged drivers.	As requested by the reviewer, we have created a graph to show graphically the results of the scoping review (Fig 3). More specifically, for each of the three domains, Fig 3 depicts visually how many papers supported each particular driver.

VERSION 2 – REVIEW

REVIEWER	Khowaja, Bakhtawar The Aga Khan University, Obstetrics and Gynecology
REVIEW RETURNED	15-Mar-2023
GENERAL COMMENTS	I am pleased with the responses and actions taken - very well done.

VERSION 2 – AUTHOR RESPONSE

BMJ Open-2022-070454 - "Behavioural factors associated with fear of litigation as a driver for the increased use of caesarean sections: A scoping review"

Dear Editor and Dear reviewers,

We are pleased that the reviewers are satisfied with the answers provided and the revisions of the manuscript. Please see below the responses to the two comments referred to by the editor.

Sincerely,

Ana Pilar Betrán on behalf of the team of co-authors

Comment	Response
Please provide an answer to Reviewer 3 comment: "The authors should also be transparent about the choice of including thesis/dissertation and grey literature for India review only."	We apologize for this mistake. This was a reporting error on our side and we have corrected it (Annex). We did not search or include grey literature or dissertations in this scoping review (neither India nor any other country).
Please revise the 'Strengths and limitations of this study' section of your manuscript (after the abstract). This section should contain up to five short bullet points, no longer than one sentence each, that relate specifically to the methods. The novelty, aims, results or expected impact of the study should not be summarised here.	The Strengths and limitations of the study have been revised as per the instructions as follows: • This is the first review to systematically explore global accounts of FoL and associated behavioural factors as drivers for the increased use of CS providing a broad overview of the evidence and identifying research gaps.• We employed a behavioural approach and the WHO behavioural principles which allowed us to classify drivers within specific domains (cognitive, social and environmental) offering clear pathways for the design of interventions to address these fears.• We developed a comprehensive search strategy and searched multiple databases but we did not search grey-literature and other informal publications.• While our interest was the understanding the drivers of medically unnecessary CS, the studies did not differentiate between necessary and unnecessary CS; hence, we reported FoL in relation to "increasing use of CS" rather than "increasing use of unnecessary CS".